# Glutamate Transporter 1 as a Novel Negative Regulator of Amyloid β

**DOI:** 10.3390/cells13191600

**Published:** 2024-09-24

**Authors:** Priyanka Sinha, Yuliia Turchyna, Shane Patrick Clancy Mitchell, Michael Sadek, Gokce Armagan, Florian Perrin, Masato Maesako, Oksana Berezovska

**Affiliations:** Alzheimer Research Unit, MassGeneral Institute for Neurodegenerative Disease, Massachusetts General Hospital, Harvard Medical School, 114, 16th Street, Charlestown, MA 02129, USA

**Keywords:** Alzheimer’s disease, presenilin 1, glutamate transporter 1, amyloid β, GLT1/PS1 interaction

## Abstract

Glutamate transporter-1 (GLT-1) dynamics are implicated in excitotoxicity and Alzheimer’s disease (AD) progression. Early stages of AD are often marked by hyperactivity and increased epileptiform activity preceding cognitive decline. Previously, we identified a direct interaction between GLT-1 and Presenilin 1 (PS1) in the brain, highlighting GLT-1 as a promising target in AD research. This study reports the significance of this interaction and uncovers a novel role of GLT-1 in modulating amyloid-beta (Aβ) production. Overexpression of GLT-1 in cells reduces the levels of Aβ40 and Aβ42 by decreasing γ-secretase activity pertinent to APP processing and induces a more “open” PS1 conformation, resulting in decreased Aβ42/40 ratio. Inhibition of the GLT-1/PS1 interaction using cell-permeable peptides produced an opposing effect on Aβ, highlighting the pivotal role of this interaction in regulating Aβ levels. These findings emphasize the potential of targeting the GLT-1/PS1 interaction as a novel therapeutic strategy for AD.

## 1. Introduction

One of the central events to Alzheimer’s disease (AD) pathology is the aberrant processing of amyloid precursor protein (APP), resulting in the generation of various Aβ peptides. Aβ released from cells can aggregate into insoluble fibrils and form amyloid plaques in the brains of AD patients. Among the various Aβ species, Aβ40 and Aβ42 are predominant, with the longer Aβ42 being particularly prone to forming highly fibrillogenic soluble oligomers that have significant detrimental effects [1]. These oligomers are highly neurotoxic, impairing synaptic plasticity, causing synaptic loss, and contributing to cognitive decline [2,3]. They also induce oxidative stress and inflammation, further damaging neurons [4]. Additionally, soluble Aβ (Aβ42 in particular) activates microglia and astrocytes, leading to chronic inflammation that worsens neuronal damage [5]. Moreover, these oligomers disrupt cellular processes such as calcium homeostasis and mitochondrial function, exacerbating neurodegeneration [6]. Thus, the elevated Aβ42/40 ratio plays a pivotal role in the AD disease pathogenesis [7]. Presenilin 1 (PS1) is a catalytic subunit of the γ-secretase complex that cleaves APP within the transmembrane domain, influencing Aβ species production through distinct “open” and “closed” conformational states of PS1 [8,9,10].

Increased epileptiform activity (seizures or subclinical manifestation) is frequently associated with AD. About 15–20% of AD patients experience seizures, and 20–40% show abnormal epileptiform discharges on EEG, both associated with severe cognitive decline [11,12,13,14,15,16]. There is increasing recognition that epilepsy can occur not just concurrently with but preceding cognitive decline [17,18,19,20]. Remarkably, PS1/2 mutation carriers exhibit epileptiform activity more often than sporadic AD patients, implicating presenilins in pathways related to epilepsy [21,22,23,24]. Furthermore, recent trials with anti-epileptic drugs have shown promise in reducing cognitive decline in AD patients with epilepsy [25], underscoring the significance of neuronal hyperexcitability in AD pathogenesis. However, a better understanding of the mechanism(s) underlying the link between AD pathology and hyperactivity is needed.

Glutamate transporter 1 (GLT-1, also known as EAAT2), a major glutamate transporter in the brain, regulates synaptic glutamate levels and is crucial for maintaining glutamate homeostasis. Dysregulation of GLT-1-mediated glutamate transport precedes amyloid plaque formation and neuronal dysfunction and accelerates neurodegeneration in AD mouse models [26,27,28,29]. GLT-1 dysfunction exacerbates cognitive deficits and accelerates disease progression in AD animal models [30,31], influencing Aβ metabolism and glutamatergic signaling pathways critical to AD pathogenesis [32,33,34,35,36]. However, how GLT-1 impacts PS1/γ-secretase and Aβ pathology remains elusive. Furthermore, we found that GLT-1 directly interacts with PS1 in the brain, bringing out its potential role in AD [37]. Understanding the dynamics of the GLT-1/PS1 interaction is essential for elucidating Aβ accumulation mechanisms and exploring GLT-1 as a potential therapeutic target for AD.

The current study investigates the intricate interplay between GLT-1, PS1/γ-secretase, and Aβ. We reveal the impacts GLT-1 and its binding to PS1 have on Aβ levels, the Aβ42/40 ratio, PS1 conformation, and γ-secretase activity, potentially guiding new therapeutic strategies in AD. Utilizing previously validated cell-permeable peptides (CPPs) to disrupt the GLT-1/PS1 interaction [38], we uncover the impact of direct binding between GLT-1 and PS1 on regulation of Aβ levels.

By unraveling the significance of the GLT-1/PS1 interaction, this study deepens our understanding of AD pathogenesis and paves the way for therapeutic interventions that might potentially target both Aβ and excitotoxicity.

## 2. Materials and Methods

### 2.1. Chemicals and Antibodies

The following reagents were used in this study: DAPT (Sigma-Aldrich, St. Louis, MO, USA), DMSO (Sigma-Aldrich, St. Louis, MO, USA), penicillin (Thermo Fisher Scientific, Waltham, MA, USA), and geneticin (Sigma-Aldrich, St. Louis, MO, USA). The following antibodies were used in this study: anti-GLT-1 (ab41621, Abcam, Cambridge, MA), anti-β-actin (Sigma-Aldrich, St. Louis, MO, USA), PS1 N-terminal (ab15456, Abcam, Cambridge, MA, USA), PS1 C-terminal (#5643, Cell Signaling Technology, Danvers, MA, USA), APP (#802802, BioLegend, San Diego, CA, USA), and GAPDH (#2118, Cell Signaling Technology, Danvers, MA, USA).

### 2.2. Cell Culture

Chinese hamster ovary (CHO) PS70 cells, kindly gifted by Selkoe lab (BWH, Boston, MA, USA), that stably express hPS1 and hAPP were cultured in Opti-MEM (Thermo Fisher Scientific, Waltham, MA, USA) enriched with 5% FBS (Atlanta Biologicals Inc., Flowery Branch, GA, USA). The medium for CHO PS70 cells was supplemented with puromycin (2.5 µg/mL) and geneticin (200 µg/mL). Cells were cultured in an incubator at 37 °C, 5% CO_2_. 

Primary cortical neurons from 14–16-day-old embryos of CD1 mice were enzymatically dissociated using papain (Worthington Biochemical Corporation, Lakewood, NJ, USA) and cultured in Neurobasal medium (Thermo Fisher Scientific, Waltham, MA, USA) enriched with 2% B27, 1% GlutaMax, and 1% penicillin/streptomycin (Thermo Fisher Scientific) at 37 °C with 5% CO_2_. All procedures involving mice were conducted in compliance with NIH guidelines for animal experimentation and were approved by the Massachusetts General Hospital Animal Care and Use Committee.

### 2.3. Expression Constructs and Transfections

Cells were transiently transfected with plasmids using Lipofectamine 3000 (Thermo Fisher Scientific, Waltham, MA, USA) according to the manufacturer’s protocol. 

The following plasmids were used for transfection: pcDNA (empty vector), GLT-1 cloned into pcDNA, and APP C99 YPet-mTurquoise-GL (C99 YT) biosensor. C99 YT (developed by [39] contains APP C99 as an immediate substrate for γ-secretase) and two fluorescent proteins, YPet and Turquoise-GL, connected by an 80 amino-acid linker that serve as FRET donor and acceptor fluorophores, respectively. 

### 2.4. ELISA for Aβ Species

Secreted Aβ levels from CHO PS70 cells were measured in the culture media using the Wako Human Amyloid (1–40) and Human Amyloid (1–42) ELISA kits according to the manufacturer’s instructions (Immuno-Biological Laboratories, Minneapolis, MN, USA). 

Primary neurons (12–14DIV) were treated with CPPs (5 μM; 2 h); the culture media was collected afterwards and concentrated five times using Amicon® Ultra Centrifugal Filter, 3 kDa MWCO (UFC9003, Sigma-Aldrich, St. Louis, MO, USA) before performing Aβ ELISA.

### 2.5. Cytotoxicity Assay

Roche cytotoxicity detection kit (Sigma-Aldrich, St. Louis, MO, USA) was used to measure lactate dehydrogenase (LDH) content according to the manufacturer’s instructions. Wallac 1420 Victor2 Multilabel Microplate Reader (PerkinElmer, Waltham, MA, USA) was used to measure the absorbance at 490 nm.

### 2.6. Western Blotting

Cells were lysed in RIPA lysis buffer (Thermo Fisher Scientific, Waltham, MA, USA) supplemented with protease and phosphatase inhibitor cocktail (Thermo Fisher Scientific, Waltham, MA, USA). Total protein concentration was determined using the Pierce BCA Protein Assay Kit (Thermo Fisher Scientific, Waltham, MA, USA) per manufacturer’s instructions. Subsequently, 30 µg of protein was loaded onto NuPAGE™ 4–12% Bis-Tris Protein gels (Thermo Fisher Scientific, Waltham, MA, USA) and transferred to nitrocellulose membranes (Thermo Fisher Scientific, Waltham, MA, USA) using the iBLOT2™ dry electroblotting system (Thermo Fisher Scientific, Waltham, MA, USA). Membranes were probed with specific primary antibodies, followed by corresponding IRDye 680 RD or 800 CW-conjugated secondary antibodies (LI-COR Biosciences, Lincoln, NE, USA). Protein bands were visualized using the LI-COR Odyssey CLx Infrared Imaging System (LI-COR Biosciences, Lincoln, NE, USA).

### 2.7. Spectral Förster Resonance Energy Transfer (FRET) Analysis of γ-Secretase Activity

CHO PS70 cells, transiently transfected with the C99 YT biosensor, were maintained in a 37 °C heating chamber with 4% CO_2_ using a Tokai-Hit STX-Co2 Digital CO_2_ Gas Mixing System (STFX model). Olympus FV3000RS Confocal Laser Scanning Microscope with a 10× objective was used for imaging. Excitation at 405 nm activated mTurquoise-GL within the biosensor, with simultaneous detection of emitted fluorescence at 470 ± 10 nm for mTurquoise-GL and 530 ± 10 nm for YPet. ImageJ software (Version 1.54) was used for background fluorescence reduction by subtracting median fluorescence intensity across entire images. Average fluorescence intensities per cell were measured in each channel. FRET efficiency, quantified as the Y/T ratio, was calculated by dividing YPet emission intensity by mTurquoise-GL emission intensity. MATLAB 2024a software was utilized for generating pseudocolored images based on Y/T ratios, facilitating spatial visualization and interpretation of γ-secretase activity via changes in FRET efficiency.

### 2.8. Immunocytochemistry

Cells cultured in 8-well chamber slides (Thermo Fisher Scientific, Waltham, MA, USA) were washed with PBS -/- (Thermo Fisher Scientific, Waltham, MA, USA), fixed in 4% paraformaldehyde (PFA), and incubated with 1.5% normal donkey serum (Jackson ImmunoResearch Labs, West Grove, PA, USA) for one hour to minimize non-specific binding. Following this, cells were incubated overnight at 4 °C with specific primary antibodies. After thorough washing, cells were exposed to corresponding Alexa Fluor 488- or Cy3-conjugated secondary antibodies for 45 minutes at room temperature. Finally, coverslips were mounted on the slides using Vectashield mounting medium (Vector Laboratories, Inc., Burlingame, CA, USA).

### 2.9. Fluorescence Lifetime Imaging Microscopy (FLIM)

The FLIM assay was conducted using an Olympus FV3000RS Confocal Laser Scanning Microscope equipped with a femtosecond-pulsed Spectra-Physics Mai Tai laser (Spectra-Physics, Milpitas, CA, USA). Two-photon excitation at 850 nm was utilized to excite fluorophores, and the lifetime of donor fluorescence was measured by employing a microchannel plate-photomultiplier tube R3809 (Hamamatsu, Bridgewater, NJ, USA) and SPC-830 time-correlated single-photon counting FLIM module (Becker & Hickl, Berlin, Germany). In SPC Image software (Version 9.0) (Becker & Hickl, Berlin, Germany), the baseline lifetime (t1) of the Alexa 488 donor fluorophore was calculated from images of cells stained with PS1 N-terminal antibody only (used as a negative control). Raw data from cells stained with PS1 N- and C-termini antibodies were fitted to multiple decay curves to determine donor fluorophore lifetime (t2) values in the presence of Cy3 acceptor fluorophore. Based on these values, FRET efficiency was calculated and expressed as a percentage [(t1 − t2) / t1] ∗ 100.

### 2.10. Statistics

GraphPad Prism 9 software (GraphPad Software, San Diego, CA, USA) was used for all statistical analyses. Normality was assessed with the D’Agostino–Pearson omnibus K2 or Kolmogorov–Smirnov tests. Data normalization was performed with either one-sample t-tests or Wilcoxon signed-rank tests relative to a control (mean of 1). Since each experiment tested a single independent factor, repeated-measures one-way ANOVA with Tukey’s or Friedman tests with Dunn’s post hoc comparisons analyzed matched experimental conditions. Significance was set at *p* < 0.05. All experiments were replicated 3–6 times. Imaging analyses used 50–100 cells per condition, with results reported as mean ± SEM or median (25th percentile; 75th percentile).

## 3. Results

### 3.1. GLT-1 Overexpression Reduces Aβ40 and Aβ42 Production

To investigate the impact of GLT-1 on APP processing, CHO PS70 cells stably expressing human APP and PS1 were transfected with either GLT-1 or pcDNA (empty vector) as a control. After 48 hours, conditioned media was collected, and levels of Aβ40 and Aβ42 were measured using ELISA. Cells treated with N-[N-(3, 5-difluorophenacetyl)-l-alanyl]-s-phenylglycinet-butyl ester (DAPT) γ-secretase inhibitor 24 hours post-transfection served as a negative control. Overexpression of GLT-1 led to a significant reduction in both Aβ40 and Aβ42 levels compared to those in pcDNA (Figure 1A,B). Quantitative analysis revealed a decrease of about 26% in Aβ40 (*p* = 0.0006) and 40% in Aβ42 (*p* = 0.0002) levels following GLT-1 overexpression. Given the significance of the Aβ42/40 ratio in AD pathogenesis [33], we checked whether GLT-1 overexpression impacts this ratio. We observed a significant decrease in the Aβ42/40 ratio following GLT-1 overexpression, as depicted in Figure 1C (48% decrease; *p* = 0.0047). To check whether GLT-1 affected APP expression or processing, Western blotting was performed to assess the levels of total APP and APP C-terminal fragments (CTFs) (Figure 1D). No significant change in total APP levels was observed (Figure 1E), while an expected accumulation of APP CTFs was detected (Figure 1F). To assess whether GLT-1 overexpression reduced Aβ production by inducing cell death, a lactate dehydrogenase (LDH) toxicity assay was conducted. Results from Appendix A indicate no significant increase in cell death upon transfection with GLT-1. Transfection efficiency was confirmed by Western blot analysis of GLT-1 expression (Appendix A). 

### 3.2. GLT-1 Promotes “Open” PS1 Conformation

Lower Aβ42/40 ratio is associated with the relaxed, “open” conformation of PS1/γ-secretase [40,41]. To explore if GLT-1 overexpression induced an “open” PS1 conformation, CHO PS70 cells were transfected with pcDNA or GLT-1 and stained with PS1 N-terminal and C-terminal antibodies followed by AF488- and Cy3 fluorescently labelled secondary antibodies, respectively (Figure 2A). PS1 conformational states were assessed by the efficiency of Förster resonance energy transfer (FRET) using fluorescence lifetime imaging microscopy (FLIM). Cells stained with AF488/PS1 N-terminal antibody only served as the FRET negative control. We detected lower FRET efficiency between the fluorescently labeled PS1 N- and C-termini in GLT-1 transfected cells (24% decrease; *p* = 0.02) compared to pcDNA-transfected cells, consistent with PS1 adopting the “open” conformation in GLT-1-expressing cells (Figure 2B). 

### 3.3. GLT-1 Overexpression Reduces APP Processing by γ-Secretase

Based on our findings of reduced Aβ production and APP CTFs’ accumulation in cells expressing GLT-1, we hypothesized that GLT-1, by binding to PS1/γ-secretase, may influence APP C99 processing. To investigate this, we employed previously characterized C99YT biosensor to assess APP C99 cleavage by γ-secretase in cells [39]. CHO PS70 cells were co-transfected with C99YT (schematic shown in Figure 3A) and either pcDNA or GLT-1, and C99 cleavage by γ-secretase was analyzed using spectral FRET. Imaging revealed a higher Y/T ratio in GLT-1-transfected cells (7% increase; *p* < 0.0001), suggesting reduced γ-secretase activity compared to that in the controls (Figure 3B). pcDNA-transfected cells treated with DAPT (γ-secretase inhibitor) served as a negative control. Data analysis indicated significant reductions in γ-secretase activity in these cells, as evident by an even higher Y/T ratio (11% increase; *p* < 0.0001).

### 3.4. Disruption of the GLT-1/PS1 Interaction Increases Aβ Production

To investigate the mechanism underlying GLT-1-mediated effects on Aβ generation, we utilized previously characterized cell-permeable peptides (CPPs) to disrupt the interaction between endogenously expressed GLT-1 and PS1 in primary neurons [38]. Specifically, the CPP with GLT-1 sequence involved in binding to PS1 was designated as “GLT-1 CPP,” with its scrambled counterpart termed “GLT-1 Scrambled.” Similarly, the CPP with PS1 sequence binding to GLT-1 was denoted as “PS1 CPP,” and its scrambled form “PS1 Scrambled.” Primary neurons (12–14 days in vitro; DIV) were treated with CPPs at a concentration of 5 µM for 2 hours, or with their respective scrambled peptides. Conditioned media were collected for Aβ estimation using ELISA, while neurons were lysed for Western blot analysis to assess full-length APP and APP CTFs. Conditioned media from neurons treated with either GLT-1 CPP or PS1 CPP exhibited elevated levels of both Aβ40 (Figure 4A and Figure 4D, respectively) and Aβ42 (Figure 4B and Figure 4E, respectively) compared to those treated with their scrambled counterparts. Furthermore, the Aβ42/40 ratio also increased after the CPPs treatment (Figure 4C,F). Western blotting data showed a reduction in APP CTFs following CPPs treatment (Figure 4G–L), indicating more APP CTFs’ cleavage concomitant with higher Aβ production after CPP treatment. 

## 4. Discussion

In this study, we found a novel role glutamate transporter-1 GLT-1 may play in the brain: negative regulation of Aβ production. Specifically, GLT-1 overexpression reduces both Aβ40 and Aβ42 levels and decreases the Aβ42/40 ratio by promoting an “open” PS1 conformation. To explore the cause of reduced Aβ, we investigated the effect of GLT-1 overexpression on γ-secretase activity and observed the accumulation of APP CTFs and increased ratiometric FRET signal, indicating reduced APP C99 processing by PS1/γ-secretase. Using CPPs to disrupt the GLT-1/PS1 interaction, we confirmed that the GLT-1/PS1 binding is necessary for the GLT-1-mediated effects on Aβ levels, implicating the relevance of the GLT-1/PS1 interaction in AD.

GLT-1 regulates glutamate homeostasis in the brain by maintaining optimal glutamate levels, thus mitigating excitotoxicity and neurodegeneration. Previous studies showed reduced Aβ deposition in animal models by utilizing transgenic or pharmacological approaches to restore GLT-1 expression [42,43], although without providing a mechanistic link between the GLT-1 and Aβ. Our study is consistent with these findings, demonstrating that GLT-1 overexpression decreases levels of both Aβ40 and Aβ42 and uncovering that the reduced PS1/γ-secretase processing of APP C99 substrate is responsible for the decrease in Aβ. It is plausible that GLT-1 binding to PS1/γ-secretase may allosterically modify its conformation (per our FLIM data, Figure 2) in a way that hinders C99 access to the catalytic core for processing, leading to reduced total Aβ generation while also shifting the proportion of cleaved Aβ40 vs. Aβ42.

The correlation between a reduced Aβ42/40 ratio and PS1 conformation has been well-established. Familial Alzheimer’s disease (fAD) PS1 mutations cause a pathogenic “closed” conformation, while gamma-secretase modulators (GSMs) that reduce the Aβ42/40 ratio induce a more relaxed, “open” PS1 conformation [4,34,44,45]. Our study revealed that GLT-1 induces a more “open” PS1 conformation, associated with a reduced Aβ42/40 ratio. This suggests that GLT-1 exerts a GSM-like allosteric effect on PS1, mitigating neurotoxic Aβ species production.

Unlike the conventional GSMs, pharmacological compounds that “stabilize” the γ-secretase/APP-CTF complex to process Aβ into shorter forms [46,47], GLT-1 presents a unique instance of a protein endogenously expressed in the brain modulating processivity of another protein. Indeed, acting as a GSM-like agent, GLT-1 might prove to be beneficial for AD patients, reducing aggregation-prone Aβ42 specie (and likely Aβ43) as well as Aβ40. This novel insight into GLT-1′s modulation of PS1 and Aβ production could lead to new therapeutic strategies targeting Aβ pathology in Alzheimer’s disease.

Interestingly, GLT-1 expression is higher in the brain of individuals with AD pathology but without dementia compared to those with AD pathology and dementia, indicating the involvement of GLT-1 in cognitive resilience [31,48,49]. Additionally, we have recently discovered that the GLT-1/PS1 interaction is reduced in sporadic AD brains [50] and with aging (unpublished data), suggesting that weakening of the GLT-1/PS1 interaction could promote Aβ deposition. Consistent with this possibility, our results indicate that disruption of the GLT-1/PS1 interaction by CPPs leads to an increase in Aβ load. These findings further support the hypothesis that enhancing GLT-1 expression and/or promoting GLT-1/PS1 interaction may be beneficial for AD patients. 

In astrocytes, GLT-1 is essential for maintaining extracellular glutamate levels by removing it from the synaptic cleft [51]. Alterations in GLT-1 levels can disrupt this balance, potentially leading to glutamate toxicity or inadequate clearance [52,53,54], which in turn might affect PS1 function since PS1 is involved in various cellular processes including synaptic function and neuroinflammation [36,55,56]. Our study reveals that GLT-1 impacts PS1-mediated Aβ production and PS1 conformation. Although most Aβ is produced in neurons, the role of PS1 in astrocytes undoubtedly extends beyond Aβ generation. GLT1-induced changes in PS1 conformation, thus, would have broader implications, as in addition to Aβ production, PS1 regulates mitochondrial metabolism and lactate secretion in astrocytes [57]. Given the potential interplay between GLT-1 and PS1, it would be important in future studies to explore how changes in GLT-1 expression may influence PS1-related processes in astrocytes.

Further future research should aim to elucidate the detailed mechanisms of the GLT-1/PS1 interaction and its effects on Aβ metabolism and glutamatergic signaling. This includes investigating how GLT-1 modulation influences these processes at a molecular level. Developing targeted modulators or “activators” that can specifically alter GLT-1 activity or stabilize its interaction with PS1 holds promise as a potential therapeutic strategy. Additionally, validating these findings across various animal models and patient-derived cell models, such as iPSCs, would be crucial to further assess their applicability and effectiveness in different genetic and environmental contexts. Such efforts could lead to novel treatment approaches and improved strategies for combating AD.

In conclusion, this study advances our understanding of the intricate interplay between GLT-1-mediated glutamatergic signaling and Aβ metabolism in AD and uncovers a new role that glutamate transporter-1 may play in the brain. By identifying the GLT-1/PS1 interaction as a potential modulator of these processes, this study opens new avenues for therapeutic interventions aimed at slowing down or halting the progression of AD pathology.

### Limitations of the Study

While the study provides valuable insights into mechanistic link between the two key players in AD implicated in glutamate homeostasis and Aβ metabolism, several questions remain. Further elucidation of the precise molecular mechanisms by which GLT-1 modulates γ-secretase activity and PS1 conformation is needed. Although blocking the PS1/GLT-1 binding by CPPs is a crucial tool supporting the conclusion of the study, unfortunately, no pharmacological agents (or genetic factors) that could stabilize the interaction are currently known/available to further test our hypothesis. In addition to being able to stabilize the interaction, in vivo studies would be necessary to validate these findings and further explore the translational potential of targeting GLT-1/PS1 interaction in AD treatment.

## Figures and Tables

**Figure 1 cells-13-01600-f001:**
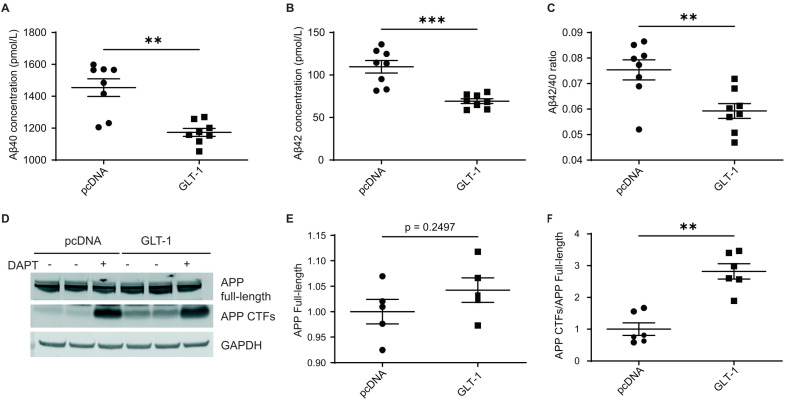
Overexpression reduces secreted Aβ in cells. Empty vector pcDNA or GLT-1 was overexpressed in CHO PS70 cells, and Aβ species were measured from the conditioned medium. Absolute concentrations of (**A**) Aβ40 and (**B**) Aβ42 are plotted in pmol/L, along with (**C**) the Aβ42/40 ratio; n = 6. Significant reduction in the above-mentioned Aβ species was observed along with a significant reduction in Aβ42/40 ratio after GLT-1 transfection as compared to those transfected with pcDNA. (**D**) Western blotting showing levels of APP full-length and CTFs in cells transfected with either pcDNA or GLT-1; n = 3. Quantification of band intensities shows (**E**) no significant change in APP full-length levels post GLT-1 transfection and (**F**) significant accumulation of APP CTFs in GLT-1-transfected cells as compared to cells transfected with pcDNA when a ratio of full-length APP CTFs/APP was performed. Statistical significance was calculated using unpaired t-test with Mann–Whitney test to compare ranks (** *p* < 0.01, *** *p* < 0.001).

**Figure 2 cells-13-01600-f002:**
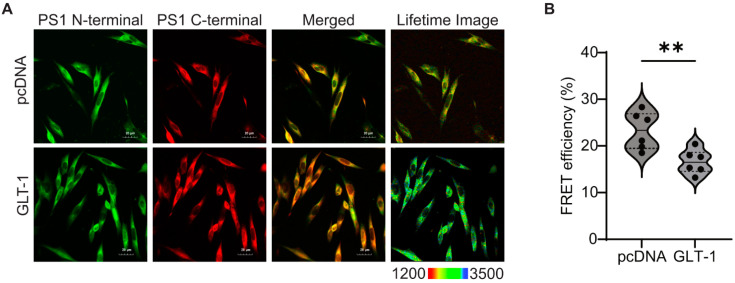
GLT-1 overexpression leads to “open” PS1 conformation. PS1 conformation was analyzed after pcDNA/GLT-1 transfection in CHO PS70 cells using FLIM. (**A**) The cells were stained with PS1 N- and C-termini antibodies followed by Alexa fluor 488 and Cy3 fluorescent antibodies, respectively. The last panel shows pseudo-colored lifetime images representing the donor fluorophore lifetime in picoseconds. The blue–green pixels represent greater distance between the fluorescently labeled PS1 N- and PS1 C-termini, indicating an “open” conformation. (**B**) The analysis of FRET efficiency was used to estimate the relative change in proximity between PS1 N- and PS1 C-termini. The graph shows the percentage of FRET efficiency, depicted as a violin plot with median (solid bars) and 25th and 75th percentiles (dotted bars); n = 6 independent experiments. Statistical significance was assessed using unpaired *t*-test with Mann–Whitney test to compare ranks (** *p* < 0.01).

**Figure 3 cells-13-01600-f003:**
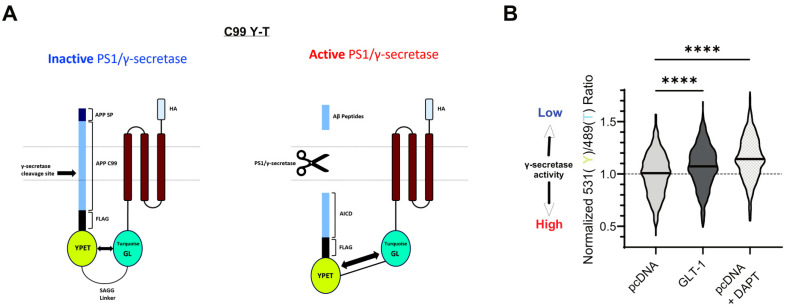
GLToverexpression reduces APP C99 processing by γ-secretase. (**A**) The schematic illustrating the structure of the C99 YT biosensor. Endogenous PS/γ-secretase cleaves the C99 portion of the biosensor, leading to a reduction in FRET between YPet (Y; donor) and Turquoise-GL (T; acceptor), and thus revealing γ-secretase activity. (**B**) The graph shows a normalized 531/489 nm ratio reflecting FRET efficiency between the Y and T fluorescent moieties, depicted as a violin plot. The higher the 531/489 nm ratio, the lower the γ-secretase activity. The median value is shown by solid bars; n = 9 independent experiments. pcDNA-transfected cells treated with DAPT (γ-secretase inhibitor) served as a positive control. Statistical significance was determined using Kruskal–Wallis ANOVA with Dunn’s multiple comparison test (**** *p* < 0.0001).

**Figure 4 cells-13-01600-f004:**
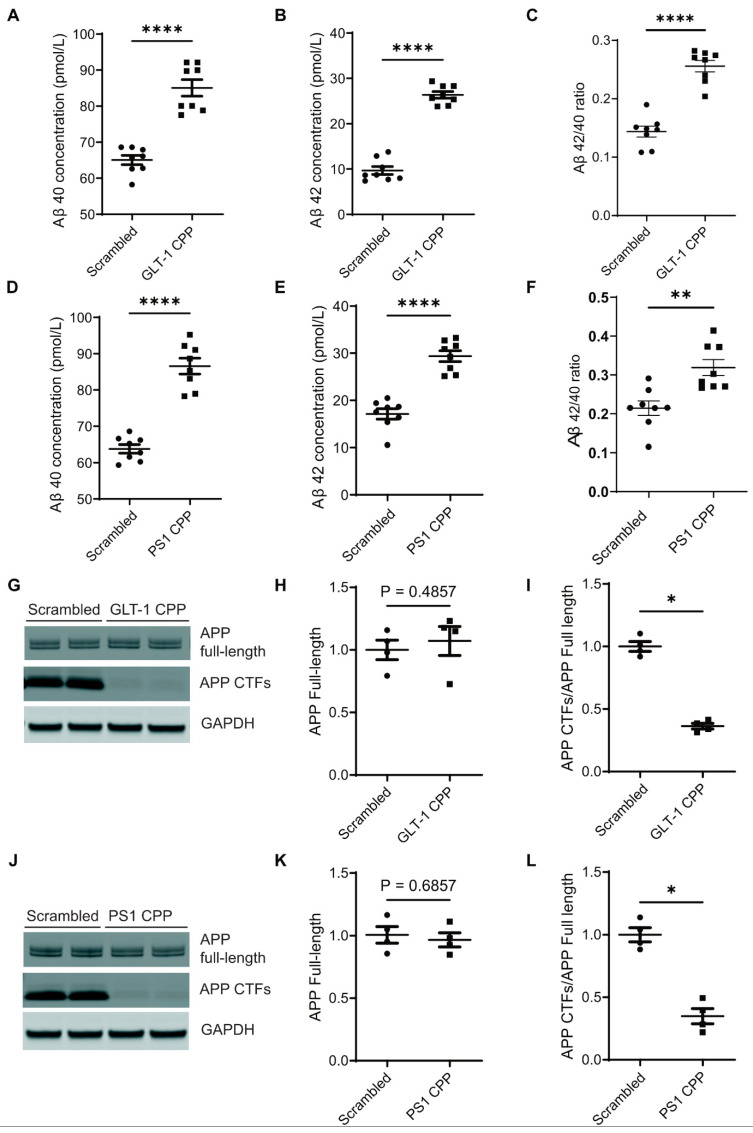
Inhibition of the GLT-1 interaction with PS1 increases secreted Aβ in primary neurons. Primary neurons (12–14 DIV) were treated with 5 µM cell-permeable peptide (CPP) for 2 h, and Aβ species in the conditioned medium were measured by ELISA. After enriching the media for secreted Aβ, absolute concentrations of Aβ40 and Aβ42 were plotted in pmol/L, along with the Aβ42/40 ratio; n = 8. (**A**) Aβ40 levels after GLT-1 CPP and (**D**) PS1 CPP treatments. Absolute concentrations of Aβ42 are shown after (**B**) GLT-1 CPP and (**E**) PS1 CPP treatment. (**C**,**F**) Aβ42/40 ratios after GLT-1 CPP and PS1 CPP treatments, respectively. (**G**,**J**) Western blotting showing levels of APP full-length and CTFs in neurons treated with either GLT-1 or PS1 CPPs or their scrambled counterparts; n = 3. (**H**,**K**) Quantification of band intensities shows no significant change in APP full-length levels post-CPP treatment. (**I**,**L**) There was significant accumulation of APP CTFs in CPP-treated neurons as compared to neurons treated with their scrambled counterparts when a ratio of full-length APP CTFs/APP was calculated. Statistical significance was calculated using unpaired t-test with Mann–Whitney test to compare ranks (* *p* < 0.05, ** *p* < 0.01, **** *p* < 0.0001).

## Data Availability

All data supporting the findings of this study are available within the paper and its Appendix A. Raw data are available from the corresponding author upon reasonable request.

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
