# Peer review of "Glutamate Transporter 1 as a Novel Negative Regulator of Amyloid β"

_cells, 2024, doi:10.3390/cells13191600_

Round 1
Reviewer 1 Report
Comments and Suggestions for Authors
In the present manuscript, the Authors would to investigate the role of GLT1/PS1 interaction on the extracellular release of beta amyloid peptides.
The Introduction well explained the background and the aim of the work. However, the author have to specify in this section what extracellular release of beta amyloid determines and its role in the AD pathogenesis. The experimental design and the adopted methodologies are well explained and clear. Results are well presented and discussion supported the obtained findings. The discussion could be improved with the future perspectives, considering also the applicability of the findings in patient-derived models, able to replicate the in vivo conditions.
Although the manuscript is well written, it lacks of references in the appropriate section in the both available versions.
Author Response
Comment 1: In the present manuscript, the Authors would to investigate the role of GLT1/PS1 interaction on the extracellular release of beta amyloid peptides. The Introduction well explained the background and the aim of the work. However, the author have to specify in this section what extracellular release of beta amyloid determines and its role in the AD pathogenesis.
Response 1: Thank you for bringing this to our attention. We have revised the introduction accordingly by adding the following:
“…Aβ released from cells can aggregate into insoluble fibrils and form amyloid plaques in the brains of AD patients. Among the various Aβ species, Aβ40 and Aβ42 are predominant, with the longer Aβ42 being particularly prone to forming highly fibrillogenic soluble oligomers that have significant detrimental effects (Selkoe, 2012). These Aβ-containing aggregates are highly neurotoxic, impairing synaptic plasticity, causing synaptic loss, and contributing to cognitive decline (Hampel et al., 2021; Zhang et al., 2023). They also induce oxidative stress and inflammation, further damaging neurons (Tamagno et al., 2021). Additionally, soluble Aβ (Aβ42 in particular) activates microglia and astrocytes, leading to chronic inflammation that worsens neuronal damage (Benedetto et al., 2022). Moreover, these aggregates disrupt cellular processes such as calcium homeostasis and mitochondrial function, exacerbating neurodegeneration (Calvo-Rodriguez et al., 2020). Thus, the elevated Aβ42/40 ratio plays a pivotal role in the AD disease pathogenesis 1
References:
- Selkoe DJ. Resolving controversies on the path to Alzheimer's therapeutics. Nat Med. 2011 Sep 7;17(9):1060-5. doi: 10.1038/nm.2460.
- Hampel H, Hardy J, Blennow K, Chen C, Perry G, Kim SH, Villemagne VL, Aisen P, Vendruscolo M, Iwatsubo T, Masters CL, Cho M, Lannfelt L, Cummings JL, Vergallo A. The Amyloid-β Pathway in Alzheimer's Disease. Mol Psychiatry. 2021 Oct;26(10):5481-5503. doi: 10.1038/s41380-021-01249-0.
- Zhang Y, Chen H, Li R, Sterling K, Song W. Amyloid β-based therapy for Alzheimer's disease: challenges, successes and future. Signal Transduct Target Ther. 2023 Jun 30;8(1):248. doi: 10.1038/s41392-023-01484-7.
- Tamagno E, Guglielmotto M, Vasciaveo V, Tabaton M. Oxidative Stress and Beta Amyloid in Alzheimer's Disease. Which Comes First: The Chicken or the Egg? Antioxidants (Basel). 2021 Sep 16;10(9):1479. doi: 10.3390/antiox10091479.
- Di Benedetto G, Burgaletto C, Bellanca CM, Munafò A, Bernardini R, Cantarella G. Role of Microglia and Astrocytes in Alzheimer's Disease: From Neuroinflammation to Ca2+ Homeostasis Dysregulation. Cells. 2022 Sep 1;11(17):2728. doi: 10.3390/cells11172728.
- Calvo-Rodriguez M, Hou SS, Snyder AC, Kharitonova EK, Russ AN, Das S, Fan Z, Muzikansky A, Garcia-Alloza M, Serrano-Pozo A, Hudry E, Bacskai BJ. Increased mitochondrial calcium levels associated with neuronal death in a mouse model of Alzheimer's disease. Nat Commun. 2020 May 1;11(1):2146. doi: 10.1038/s41467-020-16074-2.
Comment 2: The experimental design and the adopted methodologies are well explained and clear. Results are well presented and discussion supported the obtained findings. The discussion could be improved with the future perspectives, considering also the applicability of the findings in patient-derived models, able to replicate the in vivo conditions.
Response 2: Thank you for overall positive evaluation of our manuscript and for bringing our attention to future perspectives. We have now included this information in the revised manuscript:
Further, future research should aim to elucidate the detailed mechanisms of the GLT-1/PS1 interaction and its effects on Aβ metabolism and glutamatergic signaling. This includes investigating how GLT-1 modulation influences these processes at a molecular level. Developing targeted modulators or “activators” that can specifically alter GLT-1 activity or stabilize its interaction with PS1 holds promise as potential therapeutic strategy. Additionally, validating these findings across various animal models and patient-derived cell models, such as iPSC, would be crucial to further assess their applicability and effectiveness in different genetic and environmental contexts. Such efforts could lead to novel treatment approaches and improved strategies for combating AD.
Comment 3: Although the manuscript is well written, it lacks of references in the appropriate section in the both available versions.
Response 3: Thank you very much for your feedback. We have added the necessary references to the manuscript to address this issue. Please see below:
- Rothstein JD, Martin L, Levey AI, Dykes-Hoberg M, Jin L, Wu D, Nash N, Kuncl RW. Localization of neuronal and glial glutamate transporters. Neuron. 1994 Sep;13(3):713-25. doi: 10.1016/0896-6273(94)90038-8.
- Rimmele TS, Li S, Andersen JV, Westi EW, Rotenberg A, Wang J, Aldana BI, Selkoe DJ, Aoki CJ, Dulla CG, Rosenberg PA. Neuronal Loss of the Glutamate Transporter GLT-1 Promotes Excitotoxic Injury in the Hippocampus. Front Cell Neurosci. 2021 Dec 29;15:788262. doi: 10.3389/fncel.2021.788262.
- Pajarillo E, Rizor A, Lee J, Aschner M, Lee E. The role of astrocytic glutamate transporters GLT-1 and GLAST in neurological disorders: Potential targets for neurotherapeutics. Neuropharmacology. 2019 Dec 15;161:107559. doi: 10.1016/j.neuropharm.2019.03.002.
- Andersen JV, Markussen KH, Jakobsen E, Schousboe A, Waagepetersen HS, Rosenberg PA, Aldana BI. Glutamate metabolism and recycling at the excitatory synapse in health and neurodegeneration. Neuropharmacology. 2021 Sep 15;196:108719. doi: 10.1016/j.neuropharm.2021.108719.
- Zoltowska KM, Maesako M, Lushnikova I, Takeda S, Keller LJ, Skibo G, Hyman BT, Berezovska O. Dynamic presenilin 1 and synaptotagmin 1 interaction modulates exocytosis and amyloid β production. Mol Neurodegener. 2017 Feb 13;12(1):15. doi: 10.1186/s13024-017-0159-y.
- Bukke VN, Archana M, Villani R, Romano AD, Wawrzyniak A, Balawender K, Orkisz S, Beggiato S, Serviddio G, Cassano T. The Dual Role of Glutamatergic Neurotransmission in Alzheimer's Disease: From Pathophysiology to Pharmacotherapy. Int J Mol Sci. 2020 Oct 9;21(20):7452. doi: 10.3390/ijms21207452.
- Sun Y, Islam S, Michikawa M, Zou K. Presenilin: A Multi-Functional Molecule in the Pathogenesis of Alzheimer's Disease and Other Neurodegenerative Diseases. Int J Mol Sci. 2024 Feb 1;25(3):1757. doi: 10.3390/ijms25031757.
- Oksanen M, Petersen AJ, Naumenko N, Puttonen K, Lehtonen Š, Gubert Olivé M, Shakirzyanova A, Leskelä S, Sarajärvi T, Viitanen M, Rinne JO, Hiltunen M, Haapasalo A, Giniatullin R, Tavi P, Zhang SC, Kanninen KM, Hämäläinen RH, Koistinaho J. PSEN1 Mutant iPSC-Derived Model Reveals Severe Astrocyte Pathology in Alzheimer's Disease. Stem Cell Reports. 2017 Dec 12;9(6):1885-1897. doi: 10.1016/j.stemcr.2017.10.016.
- Selkoe DJ. Resolving controversies on the path to Alzheimer's therapeutics. Nat Med. 2011 Sep 7;17(9):1060-5. doi: 10.1038/nm.2460.
- Hampel H, Hardy J, Blennow K, Chen C, Perry G, Kim SH, Villemagne VL, Aisen P, Vendruscolo M, Iwatsubo T, Masters CL, Cho M, Lannfelt L, Cummings JL, Vergallo A. The Amyloid-β Pathway in Alzheimer's Disease. Mol Psychiatry. 2021 Oct;26(10):5481-5503. doi: 10.1038/s41380-021-01249-0.
- Zhang Y, Chen H, Li R, Sterling K, Song W. Amyloid β-based therapy for Alzheimer's disease: challenges, successes and future. Signal Transduct Target Ther. 2023 Jun 30;8(1):248. doi: 10.1038/s41392-023-01484-7.
- Tamagno E, Guglielmotto M, Vasciaveo V, Tabaton M. Oxidative Stress and Beta Amyloid in Alzheimer's Disease. Which Comes First: The Chicken or the Egg? Antioxidants (Basel). 2021 Sep 16;10(9):1479. doi: 10.3390/antiox10091479.
- Di Benedetto G, Burgaletto C, Bellanca CM, Munafò A, Bernardini R, Cantarella G. Role of Microglia and Astrocytes in Alzheimer's Disease: From Neuroinflammation to Ca2+ Homeostasis Dysregulation. Cells. 2022 Sep 1;11(17):2728. doi: 10.3390/cells11172728.
- Calvo-Rodriguez M, Hou SS, Snyder AC, Kharitonova EK, Russ AN, Das S, Fan Z, Muzikansky A, Garcia-Alloza M, Serrano-Pozo A, Hudry E, Bacskai BJ. Increased mitochondrial calcium levels associated with neuronal death in a mouse model of Alzheimer's disease. Nat Commun. 2020 May 1;11(1):2146. doi: 10.1038/s41467-020-16074-2.
Reviewer 2 Report
Comments and Suggestions for Authors
The authors showed that the overexpression of GLT-1 decreased Abeta cleavage by gamma-secretase in CHO PS70 cells. On the contrary, the compounds which can inhibit the interaction of GLT-1 and PS-1 strongly induced Abeta generation in primary neurons.
It is intersting that GLT-1 can affect Abeta cleavage via gamma-secretase. In this study, the authors examined cell-permeable compounds only in primary neurons. Can these compounds mitigate the effects of GLT-1 overexpression in CHO PS70 cells? The authors should confirm that.
It is widely known that GLT-1 expresses mainly in astrocytes, and PS1 also expresses in glial cells. Given that GLT-1 level can affect PS1 function, what could happen in astrocytes? The authors should discuss on this point.
Minor point: The authors should write animal species of primary neurons. Did they use rat or mouse?
Author Response
Comment 1: The authors showed that the overexpression of GLT-1 decreased Abeta cleavage by gamma-secretase in CHO PS70 cells. On the contrary, the compounds which can inhibit the interaction of GLT-1 and PS-1 strongly induced Abeta generation in primary neurons. It is intersting that GLT-1 can affect Abeta cleavage via gamma-secretase. In this study, the authors examined cell-permeable compounds only in primary neurons. Can these compounds mitigate the effects of GLT-1 overexpression in CHO PS70 cells? The authors should confirm that.
Response 1: Thank you for your comment. Employing cell lines and GLT1 overexpression was valuable initially for determining GLT-1’ potential involvement in Ab modulation. However, inhibition of the interaction between endogenously expressed proteins and assessing its impact on Ab is the most crucial approach since (a): it ensures that the observed effect was not due to overexpression “artifact”, (b): proves that it is GLT-1 binding to PS1 which is critical for this modulatory effect on Ab , and importantly, (c): it is more physiologically relevant for potential therapeutic interventions.
We do anticipate that the compounds (CPPs) will disrupt the GLT-1/PS1 interaction in cell lines overexpressing GLT-1 as they do in neurons, since these CPPs are highly specific in blocking this interaction. We do not foresee any plausible reason for differing results in cell lines.
Comment 2: It is widely known that GLT-1 expresses mainly in astrocytes, and PS1 also expresses in glial cells. Given that GLT-1 level can affect PS1 function, what could happen in astrocytes? The authors should discuss on this point.
Response 2: We agree with the reviewer that GLT-1 levels can significantly impact astrocytic PS1 function, and this interaction could have notable implications in astrocytes. Thus, we will include a discussion of these potential effects in the revised manuscript as follows:
In astrocytes, GLT-1 is essential for maintaining extracellular glutamate levels by removing it from the synaptic cleft (Rothstein et al., 1994). Alterations in GLT-1 levels can disrupt this balance, potentially leading to glutamate toxicity or inadequate clearance (Rimmele et al., 2021; Pajarillo et al., 2019; Anderson et al., 2021), which in turn might affect PS1 function since PS1 is involved in various cellular processes including synaptic function and neuroinflammation (Zoltowska et al., 2017; Bukke et al., 2020; Sun et al., 2024). Our study reveals that GLT-1 impacts PS1-mediated Aβ production and PS1 conformation. Although most Aβ is produced in neurons, the role of PS1 in astrocytes undoubtedly extends beyond Aβ generation. GLT1-induced changes in PS1 conformation, thus, would have broader implications, as in addition to Aβ production, PS1 regulates mitochondrial metabolism and lactate secretion in astrocytes (Oksanen et al., 2017). Given the potential interplay between GLT-1 and PS1, it would be important in future studies to explore how changes in GLT-1 expression may influence PS1-related processes in astrocytes.
References:
- Rothstein JD, Martin L, Levey AI, Dykes-Hoberg M, Jin L, Wu D, Nash N, Kuncl RW. Localization of neuronal and glial glutamate transporters. Neuron. 1994 Sep;13(3):713-25. doi: 10.1016/0896-6273(94)90038-8.
- Rimmele TS, Li S, Andersen JV, Westi EW, Rotenberg A, Wang J, Aldana BI, Selkoe DJ, Aoki CJ, Dulla CG, Rosenberg PA. Neuronal Loss of the Glutamate Transporter GLT-1 Promotes Excitotoxic Injury in the Hippocampus. Front Cell Neurosci. 2021 Dec 29;15:788262. doi: 10.3389/fncel.2021.788262.
- Pajarillo E, Rizor A, Lee J, Aschner M, Lee E. The role of astrocytic glutamate transporters GLT-1 and GLAST in neurological disorders: Potential targets for neurotherapeutics. Neuropharmacology. 2019 Dec 15;161:107559. doi: 10.1016/j.neuropharm.2019.03.002.
- Andersen JV, Markussen KH, Jakobsen E, Schousboe A, Waagepetersen HS, Rosenberg PA, Aldana BI. Glutamate metabolism and recycling at the excitatory synapse in health and neurodegeneration. Neuropharmacology. 2021 Sep 15;196:108719. doi: 10.1016/j.neuropharm.2021.108719.
- Zoltowska KM, Maesako M, Lushnikova I, Takeda S, Keller LJ, Skibo G, Hyman BT, Berezovska O. Dynamic presenilin 1 and synaptotagmin 1 interaction modulates exocytosis and amyloid β production. Mol Neurodegener. 2017 Feb 13;12(1):15. doi: 10.1186/s13024-017-0159-y.
- Bukke VN, Archana M, Villani R, Romano AD, Wawrzyniak A, Balawender K, Orkisz S, Beggiato S, Serviddio G, Cassano T. The Dual Role of Glutamatergic Neurotransmission in Alzheimer's Disease: From Pathophysiology to Pharmacotherapy. Int J Mol Sci. 2020 Oct 9;21(20):7452. doi: 10.3390/ijms21207452.
- Sun Y, Islam S, Michikawa M, Zou K. Presenilin: A Multi-Functional Molecule in the Pathogenesis of Alzheimer's Disease and Other Neurodegenerative Diseases. Int J Mol Sci. 2024 Feb 1;25(3):1757. doi: 10.3390/ijms25031757.
- Oksanen M, Petersen AJ, Naumenko N, Puttonen K, Lehtonen Š, Gubert Olivé M, Shakirzyanova A, Leskelä S, Sarajärvi T, Viitanen M, Rinne JO, Hiltunen M, Haapasalo A, Giniatullin R, Tavi P, Zhang SC, Kanninen KM, Hämäläinen RH, Koistinaho J. PSEN1 Mutant iPSC-Derived Model Reveals Severe Astrocyte Pathology in Alzheimer's Disease. Stem Cell Reports. 2017 Dec 12;9(6):1885-1897. doi: 10.1016/j.stemcr.2017.10.016.
Comment 3: The authors should write animal species of primary neurons. Did they use rat or mouse?
Response 3: I apologize for the oversight. We used primary neurons derived from CD1 mouse in our study. We will make sure to include this information explicitly in the revised manuscript for clarity.